# LLM Agent Communication Protocol (LACP) Requires Urgent Standardization: A Telecom-Inspired Protocol is Necessary

## Abstract

This position paper argues that **the LLM agent field must urgently adopt a unified, telecom-inspired communication protocol, exemplified by our proposed LLM-Agent Communication Protocol (LACP), to overcome critical deficiencies in current ad-hoc approaches that threaten safety, interoperability, and scientific progress.** The prevailing landscape of fragmented protocols, perilously echoing early networking's "protocol wars", severely curtails agent collaboration and reliability. Our analysis identifies fundamental flaws including crippling interoperability gaps that lead to scientific stagnation, inherent insecurity due to security being an afterthought, and a lack of transactional integrity stemming from monolithic designs unsuited for critical operations.

Drawing direct inspiration from telecommunications' transformative standardization, which championed principles like layered abstraction, security by construction, minimal core with extensibility, and consensus-driven interoperability, we propose LACP. LACP is a principled, three-layer framework designed to ensure agents communicate with clear semantic intent, engage in reliable, verifiable transactions, and benefit from inherent security. It embodies its core tenets—minimal core, layered design, security by default, and content agnosticism—to provide a robust and adaptable communication foundation. We urge the NeurIPS community to spearhead the adoption of such a principled approach before current fragmentation becomes an irreversible impediment to trustworthy AI, particularly in high-stakes domains. This strategic shift is vital for unlocking the full scientific and societal potential of collaborative AI.

## 1 Introduction

The rapid proliferation of Large Language Model (LLM) agents promises to revolutionize domains ranging from healthcare and transportation to complex scientific discovery and enterprise automation [Guo et al., 2024, Fan et al., 2025, Wu et al., 2025]. These agents, capable of intricate reasoning, planning, and tool utilization, are increasingly poised to tackle real-world tasks of significant complexity. However, a critical and escalating challenge threatens to derail this trajectory: the fragmented and ad-hoc nature of inter-agent communication. As more LLM agents are deployed, the lack of standardized protocols makes it difficult for them to work together, scale effectively, or be verified for safety and reliability [Yang et al., 2025].

The current landscape reveals a patchwork of emerging protocols—such as Model Context Protocol (MCP) for tool invocation, Agent-to-Agent Protocol (A2A) for peer-like task outsourcing, Agent Network Protocol (ANP) for decentralized discovery, and the Agent Communication Protocol

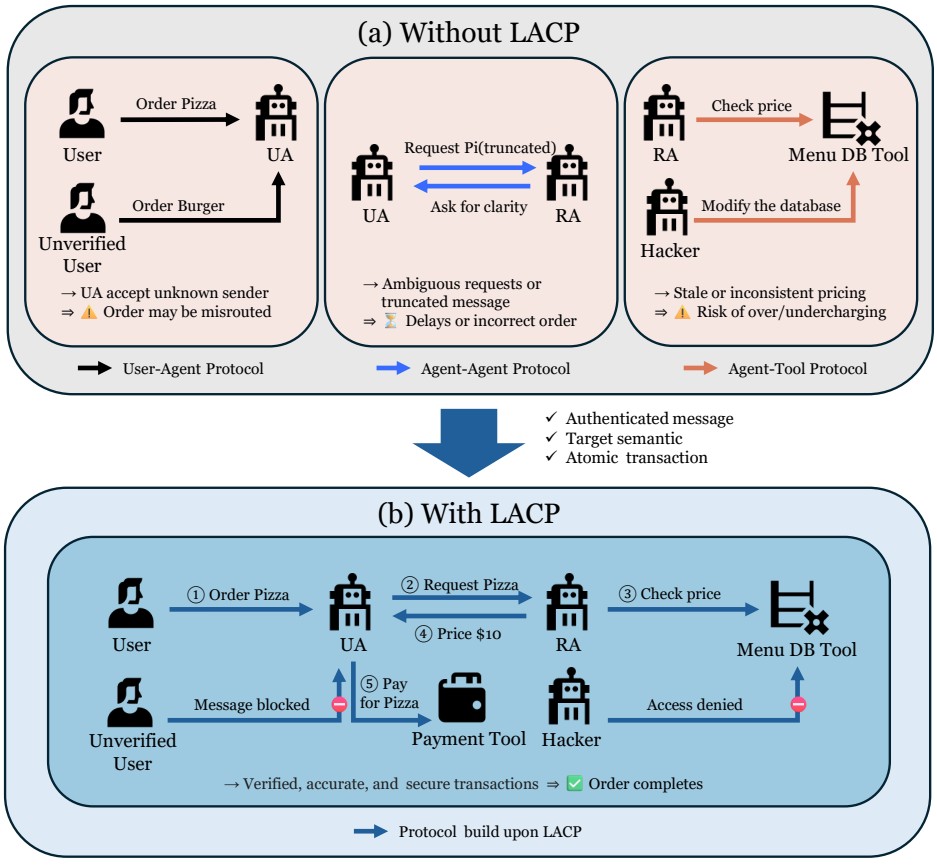

Figure 1: **LLM Agent Communication: The Chaos of Ad-Hoc Protocols vs. The Clarity of LACP.**
*(a) Without LACP: Fragmentation, ambiguity, and failure.* Ad-hoc protocols lack authentication and semantic alignment, leading to disrupted workflows, miscommunication, and unauthorized operations. Messages may be malformed or incomplete, escalating coordination cost and risking system inconsistency.
*(b) With LACP: Structured, secure, and transactional.* LACP ensures each message is **authenticated**, semantically grounded in a clear **target**, and executed as part of an **atomic transaction**. This enforces end-to-end integrity, minimizes ambiguity, and enables reliable multi-agent collaboration in safety-critical tasks.

(ACP) for REST-native messaging—each addressing distinct interoperability tiers or deployment contexts [Ehtesham et al., 2025]. While these efforts, alongside others like OpenAI's Function Calling [OpenAI, 2023] and LangChain's Agent Protocol [LangChain, 2024], represent valuable steps, they often cater to specific interaction paradigms or ecosystems. As highlighted by recent surveys [Yang et al., 2025, Ehtesham et al., 2025], they have yet to converge on a comprehensive, universally adopted framework that ensures robust, secure, and transactional communication across heterogeneous systems. This "communication chasm" actively impedes the transition from promising research prototypes to dependable, mission-critical applications and limits the formation of a truly "connected network of intelligence". The chaotic nature of current ad-hoc communication and its detrimental effects are illustrated in Figure 1 (a), showcasing how ambiguities, security vulnerabilities like unverified user interactions, and unreliable information truncatio frequently lead to task failures. In stark contrast, Figure 1 (b) demonstrates how a standardized, principled framework like LACP can bring clarity, security, and reliability, ensuring successful multi-agent operations.

This situation perilously mirrors the "protocol wars" of early computer networking (1970s-1990s) [Tanenbaum, 2003, Leiner et al., 2009], where a plethora of competing proprietary stan-

dards (e.g., IBM's SNA, Digital's DECnet, Xerox's XNS) fractured the digital landscape and stalled innovation until the adoption of TCP/IP. The historical lesson is unequivocal: without a common communication foundation, complex distributed systems cannot achieve their full potential or be deployed with the requisite confidence and safety assurances. The challenge is compounded by what can be termed an "Agent Communication Trilemma"—the difficulty in simultaneously optimizing for versatility, efficiency, and portability in agent communication [Marro et al., 2024]—a problem that ad-hoc solutions or even flexible meta-protocols like Agora [Marro et al., 2024] may not fully resolve when foundational safety and transactional guarantees are paramount for high-stakes environments [Yang et al., 2024].

Drawing direct inspiration from telecom's transformative standardization—particularly its adoption of layered architectures, robust interoperability protocols, and embedded security-by-design—this paper argues for the imperative of a unified, telecom-inspired LACP. Current ad-hoc methodologies are fundamentally deficient, leading to crippling interoperability gaps, systemic insecurity, and a lack of transactional integrity, especially in safety-critical operations. These issues collectively hinder scientific progress and the reliable deployment of advanced multi-agent systems.

**Position: To ensure the safe and reliable of multi-agent LLM systems, particularly in physical or safety-critical domains, the NeurIPS community must spearhead the immediate collaborative development and adoption of a unified, telecom-inspired communication protocol. Such leadership is crucial not only to mitigate risks stemming from protocol fragmentation but also to unlock the full scientific and societal potential of collaborative AI.**

To substantiate this position, this paper offers the following contributions:

- An analytical evaluation of extant agent communication protocols (Section 2), identifying critical deficiencies and systemic risks informed by established networking principles and contemporary surveys.

- A distillation of essential design insights from telecom standardization history (Section 3), adapted to address current challenges in LLM agent communication and mitigate past pitfalls.

- The introduction and exposition of LACP (Section 4), detailing its core principles, three-layer architecture, theoretical advantages, and illustrative application scenarios.

- A proactive engagement with potential counterarguments to standardization (Section 5) and the delineation of a clear, actionable path forward for collaborative protocol development and adoption.

By integrating insights from telecommunications protocol engineering with cutting-edge LLM safety research, LACP offers a forward-compatible foundation for increasingly autonomous multi-agent systems, rather than a mere short-term remediation. The cost of continued inaction is stark: entrenched fragmentation, compromised operational safety, and significantly impeded scientific progress. Conversely, the timely adoption of a common protocol promises to catalyze a new era of collaborative AI capabilities, unlocking applications presently beyond the reach of today's fragmented ecosystem.

## 2 The Current State of LLM-Agent Communication

The landscape of LLM agent communication is currently characterized by a fragmented array of protocols, predominantly developed by leading AI companies and organizations, each designed to address specific interoperability tiers or deployment contexts [Ehtesham et al., 2025]. Each presents different approaches and inherent limitations (summarized in Table 1), creating significant challenges for developers, researchers, and organizations striving to build reliable, interoperable, and scientifically verifiable agent systems [Yang et al., 2025, Ehtesham et al., 2025].

### 2.1 Landscape of Existing Protocols

The contemporary environment is characterized by a multitude of communication protocols, typically originating from various AI development entities and research consortia to address specific operational requirements or ecosystem niches. Notable examples include:

Table 1: Comparison of agent communication protocols and how LACP addresses their limitations.

| Framework | Released | Developer | Interface Type | Key Features | Security Features |
|---|---|---|---|---|---|
| OpenAI Functions | June 2023 | OpenAI | JSON schema | Single-step tool calling | API key auth only |
| Agent Protocol | Nov 2024 | LangChain | REST API | Framework-agnostic APIs | HTTP/JWT auth only |
| MCP | Nov 2024 | Anthropic | JSON-RPC/HTTP | Tool-resource-prompt, Context std. | OAuth 2.1; Access controls |
| ACP | 2024 (Draft) | IBM/LF | JSON-RPC | Multimodal, Async streaming | Signed capability tokens; RBAC bridge |
| ANP | Mar 2024 | Community | DID/JSON-LD | Decentralized identity, Discovery | W3C DIDs, Encrypted comms |
| Agora | Oct 2024 | Academia | Meta protocol | Hybrid NL/structured, Negotiation | Hash-based ID, Security PDs |
| Agent2Agent (A2A) | Apr 2025 | Google | HTTP/Protobuf | Peer-to-peer, Agent Cards, Async | Capability discovery, (Auth/TLS) |
| **LACP (Proposed)** | **2025** | **Open Standard** | **Layered** | **Layered semantics, transactions** | **E2E crypto, 2PC, Auth (core)** |

- **OpenAI Functions** [OpenAI, 2023]: Introduced in June 2023, this mechanism lets LLMs invoke predefined tools via a JSON schema—ideal for single-step, model-to-tool calls.

- **Agent Protocol** [LangChain, 2024]: Released by LangChain in Nov 2024, it defines an *OpenAPI-based HTTP interface* that is framework-agnostic, enabling agents to expose/consume actions through standard endpoints.

- **Model Context Protocol (MCP)** [Anthropic, 2024]: Announced by Anthropic in Nov 2024, MCP uses JSON-RPC over HTTP plus OAuth 2.1–signed requests to allow secure tool invocation and typed data exchange.

- **Agent Communication Protocol (ACP)** [BM BeeAI, 2024]: A 2024 Linux-Foundation draft, ACP specifies a REST-native, multi-part messaging layer with multimodal payloads, asynchronous streaming, and capability-token security.

- **Agent Network Protocol (ANP)** [Agent Network Protocol Contributors, 2024]: A community project (Mar 2024) that employs W3C Decentralized Identifiers (DIDs) and JSON-LD graphs for open-internet agent discovery and encrypted collaboration.

- **Agora** [Marro et al., 2024]: An academic meta-protocol (Oct 2024) that lets agents dynamically negotiate communication by blending natural-language "routine drafts" with structured JSON messages.

- **Agent2Agent (A2A)** [Google, 2025]: Released by Google in Apr 2025, A2A enables peer-to-peer agent communication over HTTP/Protobuf and introduces capability-centric *Agent Cards* for enterprise-scale workflows.

## 2.2 Systemic Risks and Critical Deficiencies

The extant patchwork of LLM agent communication protocols exposes several critical deficiencies and systemic risks that threaten the advancement and safe deployment of multi-agent systems:

- **Fragmented Ecosystem.** A jumble of incompatible formats and ad-hoc APIs forces developers to build brittle middleware just to get agents talking. This fragmentation not only hinders real-world deployments but also makes rigorous, reproducible research nearly impossible, leading to stagnation in benchmarking and evaluation.

- **Security as an Afterthought.** Most existing protocols lack built-in authentication, integrity checks, or encryption. By treating security as optional, they leave agents wide open to tampering, spoofing, and adversarial attacks—risks that only grow when reasoning is offloaded to large, opaque models.

- **Monolithic Design and Operational Fragility.** Without clear layering or modular boundaries, many agent frameworks turn into tangled, monolithic stacks. This complexity makes debugging and maintenance a nightmare and prevents systems from scaling or evolving gracefully when new requirements emerge.

- **Ill-Defined Core Functionality.** In place of a lean "narrow waist" of essential primitives, current approaches either over-specify peripheral features or leave core operations vague. The result is redundant reimplementation of basic capabilities and slowed innovation at the application layer.

Table 2: Evolution of wireless protocols and their broader impact

| Generation | Key Innovation(s) | Protocol(s) | Impact |
|---|---|---|---|
| 1G | Analog voice, basic mobility | AMPS, NMT | Cellular access to mobile telephony |
| 2G | Digital modulation, encryption | GSM | Secure, global-scale mobile voice and SMS |
| 3G | Soft handoff, mobility management | WCDMA, CDMA2000 | Mobile internet, video telephony |
| 4G | All-IP stack, bearer abstraction | LTE | Streaming, app ecosystems, mobile cloud |
| 5G | Service-based architecture | NR, HTTP/2, JSON | Massive IoT, URLLC, network slicing |
| 6G (planned) | Distributed intelligence, sensing | IMT-2030 | AI-native communication, holographic presence |

# 3 Insights from Telecommunications Standardization

The historical trajectory of telecommunications, from nascent fragmented systems to globally harmonized standards, offers profound analogies and instructive lessons for addressing the current challenges in LLM agent communication [Andrews et al., 2014]. The telecommunications industry successfully navigated and resolved many of the same issues now emerging in multi-agent AI systems, including fragmented protocols, lack of semantic alignment, security vulnerabilities, and limited interoperability.

## 3.1 Historical Evolution and Relevance

The telecommunications industry's progression from fragmented analog systems to unified global standards, such as those enabling GSM's digital encryption, authentication, and international interoperability, provides instructive parallels for LLM agent communications. Early wireless communication was marked by isolated, point-to-point channels with idiosyncratic message formats and largely unregulated spectrum usage. A pivotal inflection point occurred in 1906 with the International Radiotelegraph Convention, which, though narrow in technical scope, introduced standardized station identifiers and a universal distress call (SOS). This event critically established the principle that transnational coordination was both feasible and beneficial, paving the way for future international standardization bodies like the International Telecommunication Union (ITU) [ITU, 1865] and the 3rd Generation Partnership Project (3GPP) [3GPP, 1998].

This historical trajectory—uncoordinated early innovation followed by structured protocolization—and the progressive layering and standardization observed in telecom protocols (summarized in Table 2) offer both a historical analogue and a design blueprint for LLM agent communication. A detailed technical chronology is provided in Appendix A.

## 3.2 Core Principles for Protocol Design

Drawing from the telecommunications industry's successful standardization experience, we propose four foundational principles essential for the robust design of future LLM-agent communication protocols, directly addressing the previously identified deficiencies:

- **Consensus-Driven Open Standards for Interoperability.** Telecom achieved global reach by building protocols through open, collaborative bodies (e.g., ITU, 3GPP). A similar, community-led process can unify LLM agent interfaces, eliminate fragmentation, and ensure that new research builds on a stable, shared foundation.

- **Security by Construction.** Mature telecom stacks bake in cryptographic identity, integrity, and confidentiality at every layer. By making security non-negotiable, LLM agent protocols can protect against tampering and adversarial threats from day one.

- **Layered Abstractions for Modularity.** Just as the OSI model cleanly separates concerns (physical, link, network, etc.), a layered agent stack lets teams evolve individual components—transaction management, message routing, error recovery—without destabilizing the whole system.

- **Minimal Core, Extensible Edge.** Effective telecom standards define a slim set of universal primitives (the "narrow waist") while providing clear hooks for domain-specific extensions. This balance ensures everyone agrees on the essentials while empowering innovation at the edges.

Table 3: Feature support in existing agent-communication proposals.

| Feature | Functions call | Agent Protocol | MCP | A2A | Agora | LACP |
|---|---|---|---|---|---|---|
| Cross-framework interoperability | ○ | ● | ● | ● | ● | ● |
| Multi-agent coordination | × | ○ | ○ | ● | ● | ● |
| Layered architecture | × | × | ● | ○ | ○ | ● |
| End-to-end message signing | × | × | ○ | ● | ○ | ● |
| Transaction guarantees | × | ○ | × | ○ | ● | ● |
| Retry/timeout mechanisms | × | ○ | ○ | ○ | ○ | ● |
| Independence from specific LLM | × | ● | ● | ● | ● | ● |

×: Not supported, ○: Partially supported, ●: Supported

Table 4: Core LACP message types (all wrapped in a JWS envelope).

| Type | Mandatory fields | Optional fields | Purpose |
|---|---|---|---|
| PLAN | intent_id, role, natural_language | graph_ops | express high-level intent |
| ACT | intent_id, tool_call, params | deadline, cost_cap | invoke an external tool |
| OBSERVE | intent_id, status, output | metrics | return results / status |

## 4 LACP: A Principled Framework for LLM-Agent Communication

The systemic deficiencies inherent to contemporary LLM-agent communication paradigms necessitate a foundational shift toward standardization. Inspired by proven telecom engineering principles, we introduce the LACP—a principled framework that ensures secure, reliable, and interoperable communications within multi-agent systems.

### 4.1 Position Statement Reiterated

The analysis of systemic risks (Section 2) and historical insights (Section 3) highlights an urgent imperative: achieving reliability, security, and scientific rigor in multi-agent LLM systems, particularly those interacting with real-world environments, fundamentally requires adopting a telecom-inspired communication standard. LACP is explicitly designed to address this imperative, promoting agent safety, scientific reproducibility, and accelerating the growth of the entire multi-agent ecosystem.

Contemporary agent communication protocols, while addressing specific use cases, fail to simultaneously deliver the four pillars essential for production-grade multi-agent systems: comprehensive cross-framework interoperability, explicit multi-agent coordination, mandatory cryptographic security, and robust transaction guarantees. Table 3 demonstrates this critical limitation—no existing protocol provides comprehensive support across all dimensions, creating systematic vulnerabilities that compound in safety-critical deployments.

LACP directly addresses this gap through a principled architectural approach that treats security, reliability, and interoperability as foundational design constraints rather than optional extensions. By drawing from telecommunications' proven layered protocol design, LACP implements a three-tier architecture where each layer addresses distinct concerns while maintaining clean separation of responsibilities, enabling both robust operation and independent evolution.

### 4.2 Three-Layer Architecture: Separation of Concerns

LACP's architecture (Figure 2) implements three mutually-insulated layers, each with well-defined interfaces that enable independent evolution while ensuring system-wide coherence:

- **Semantic Layer: Intent Expression and Clarity.** This layer is responsible for conveying the communicative intent between agents. It defines a minimal set of fundamental message types (e.g., PLAN, ACT, OBSERVE, ERROR) representing common inter-agent actions. While the specific content and structure of these semantic messages can be domain-specific to allow for rich and varied interactions (supporting diverse application scenarios), they are framed within this commonly understood set of intents, ensuring clarity of purpose. This minimal "narrow waist" approach ensures universal interoperability while providing clear extension points for domain-specific semantics—complex workflows and specialized

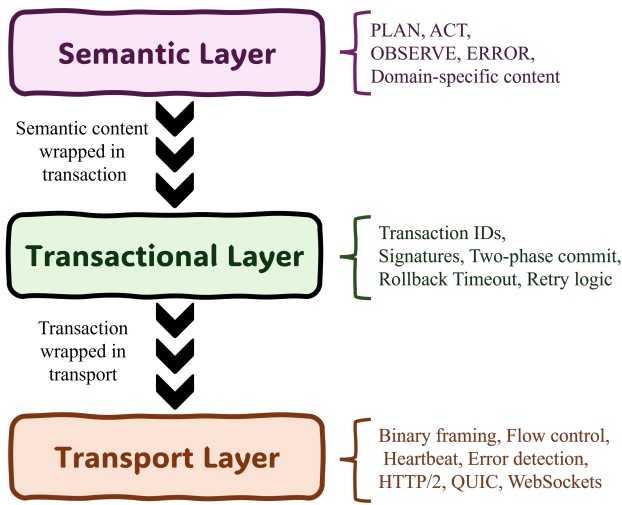

Figure 2: LACP's three-layer architecture. Independent layers enable secure, reliable communication with clearly defined semantic intents, transactional integrity, and flexible transport mechanisms.

data structures can be embedded within these standardized message envelopes without compromising cross-system compatibility. Table 4 presents three minimal core message set mandated for all implementations.

- **Transactional Layer: Reliability and Integrity Guarantees.** This layer ensures the reliability and integrity of multi-message interactions and complex workflows. It provides indispensable mechanisms such as unique transaction identifiers for idempotency (preventing duplicate processing), message sequencing, support for atomic operations across multiple agents (e.g., via two-phase commit concepts or sagas), timeout management, and configurable retry logic. Critically, this layer mandates cryptographic signatures for all messages, ensuring authenticity and integrity, which are paramount for secure operations, addressing a key need for robust interactions.

- **Transport Layer: Efficient and Secure Message Delivery.** The transport layer handles secure, efficient message carriage across diverse network infrastructures while abstracting underlying transport mechanisms. This enables LACP to operate seamlessly across different deployment environments—from local inter-process communication to global distributed systems—while maintaining consistent performance and security characteristics. The layer supports multiple transport protocols (HTTP/2, QUIC, WebSockets) chosen dynamically based on deployment requirements, with efficient binary message framing, network flow control, and heartbeat mechanisms for connection liveness monitoring. This transport agnosticism ensures LACP can evolve with networking technologies while maintaining backward compatibility.

This layered design cleanly separates concerns—semantics, transactional integrity, and transport—so that intent is explicit, safety-critical guarantees are enforced, and messages ride on efficient, secure carriers while each layer remains free to evolve. Figure 3 makes the flow concrete: a semantic PLAN payload is wrapped by a signed, two-phase-commit envelope and then by a binary transport frame, yielding a secure, reliable, and interoperable message path that scales to complex multi-agent systems.

### 4.3 Theoretical Advantages and Anticipated Impact

LACP's principled design and layered architecture offer significant theoretical advantages, poised to catalyze transformative impacts across the LLM agent ecosystem:

- **Enhanced Safety and Reliability:** By embedding security-by-default and a robust transactional layer, LACP provides a verifiable foundation for agent actions. This directly mitigates

```
// (1) Semantic layer
{
    "type": "plan",
    "intent_id": "move_object_plan_1",
    "role": "planner-agent",
    "natural_language": "Move object from A to B",
    "graph_ops": [
        { "op": "grasp",   "args": {"from": "A"} };
        { "op": "move",    "args": {"to":   "B"} };
        { "op": "release" }
    ]
}

// (2) Transactional layer
{
    "transaction_id": "tx-28a9ef",
    "sequence_num": 42,
    "timestamp": "2025-05-13T10:15:30Z",
    "source_agent": "planner-agent",
    "target_agent": "robot-controller",
    "payload": /* semantic object above */,
    "signature": "<base64url-ed25519>",
    "timeout_ms": 5000
}

// (3) Transport frame header
[0x00 0x2F ...]
<transactional message as binary>
[0xAD 0xDE]              // frame checksum
```

Figure 3: Layer-by-layer encoding of a PLAN message in LACP. (1) the bare semantic payload, (2) the same payload wrapped by the transactional layer with a JSON Web Signature, and (3) the truncated binary transport frame.

risks of erroneous or malicious operations, especially in safety-critical domains where current protocols fall short.

- **System-Wide Interoperability and Composability:** LACP aims to dismantle the "communication chasm" by providing a universal standard. This allows agents from diverse origins to seamlessly connect and specialized services to be modularly integrated, fostering a richer, more capable multi-agent ecosystem than the current fragmented state.

- **Accelerated Scientific Progress and Innovation:** A common communication protocol is indispensable for scientific rigor. LACP offers a stable baseline for reproducible experiments and fair benchmarking, addressing the "benchmark rot". By standardizing foundational communication, it liberates researchers to focus on advancing core AI capabilities and innovating at the application layer, rather than on bespoke integrations.

- **Facilitation of a Dynamic Agent Economy:** With a trusted and standardized means of interaction, LACP can underpin a more sophisticated agent economy. Agents could more reliably discover, contract, and collaborate on complex tasks, unlocking new economic and societal applications currently unfeasible.

- **Future-Proof Scalability and Evolvability:** The modular, layered design and the minimal core with well-defined extension points ensure LACP can adapt to new agent capabilities and evolving communication paradigms without requiring disruptive overhauls, promising long-term relevance and scalability.

## 5 Alternative Views & Rebuttals

The proposal to standardize LLM agent communication via LACP invites critical assessment. We proactively address anticipated objections:

**Objection 1: Standardization will stifle innovation.**
*Rebuttal:* LACP standardizes foundational communication primitives—syntax, transactional integrity, security—not core agent intelligence or learning algorithms. Analogous to TCP/IP's role in fostering internet innovation, LACP aims to provide a stable, interoperable substrate. By abstracting essential reliability and security concerns to the protocol level, LACP liberates resources for innovation in

higher-order agent functionalities and application-specific logic, consistent with the need for protocols that enable, rather than restrict, capability evolution.

**Objection 2: The semantic diversity of agent tasks precludes a unified grammar.**
*Rebuttal:* LACP employs the "narrow waist" architectural principle, analogous to the internet protocol suite. It standardizes a minimal set of essential message types (e.g., PLAN, ACT, OBSERVE) and interaction patterns, while deliberately remaining agnostic to payload content. This design allows diverse domain-specific semantics and complex data structures to be embedded within standardized message envelopes, thereby balancing interoperability with the requisite flexibility for varied application scenarios.

**Objection 3: Additional protocol overhead will degrade performance, particularly in latency-sensitive applications.**
*Rebuttal:* LACP's design inherently considers performance, a critical evaluation dimension for agent protocols. The use of efficient binary encodings, modern transport protocols (e.g., QUIC, HTTP/2), and header compression techniques can substantially mitigate transmission overhead. We project that the overhead attributable to LACP's transactional and security features will be marginal (estimated at <5% for typical message sizes) relative to the significant improvements in communication reliability, security, and interoperability. This modest performance cost is particularly justified in safety-critical applications where correctness and verifiability are paramount.

**Objection 4: Existing agent frameworks and their proprietary protocols offer adequate communication solutions.**
*Rebuttal:* Current frameworks, while valuable, often present ecosystem-specific or incomplete communication mechanisms, as detailed in Table 1 and Section 2. They frequently lack comprehensive end-to-end security, robust transactional integrity for complex multi-step operations, or true cross-framework interoperability without substantial custom integration. LACP is conceptualized to address these identified critical deficiencies, potentially integrating as a foundational layer to augment, rather than merely replace, existing systems.

# 6 Conclusion

We stand at a critical juncture in multi-agent AI development. The current fragmentation of LLM agent communication protocols—echoing the costly "protocol wars" that once paralyzed computer networking—threatens the field's scientific integrity and deployment in safety-critical domains. Without immediate standardization, we risk architecting collaborative AI on fundamentally unstable foundations.

Our analysis demonstrates that existing protocols systematically fail to provide cross-framework interoperability, robust multi-agent coordination, mandatory cryptographic security, and atomic transaction guarantees simultaneously. LACP addresses this gap through a principled, telecom-inspired architecture that treats these capabilities as foundational requirements rather than optional features.

**The window for action is rapidly closing.** As LLM agents proliferate across industries—from autonomous vehicles to medical diagnostics—incompatible communication patterns will become increasingly entrenched and difficult to reverse. The telecommunications industry's transformation offers both blueprint and warning: standardization delayed is innovation denied.

We call upon the NeurIPS community to lead coordinated standardization through three pillars: (1) **Technical Foundation**—open-source LACP implementations and comprehensive interoperability failure datasets; (2) **Rigorous Process**—multi-stakeholder working groups with transparent, consensus-driven specification development; (3) **Strategic Engagement**—pilot deployments across diverse domains with domain-specific extensions preserving core interoperability.

LACP lays the groundwork for multi-agent systems that are verifiable, secure, and scientifically reproducible. Acting swiftly ensures that collaborative AI is built on trust and universal interoperability rather than on brittle, ad-hoc fixes. The choice is clear: adopt a principled standard that unlocks transformative potential, or continue paying the rising costs of fragmentation. Embracing LACP is therefore pivotal to realising the "collective-intelligence infrastructure" of the future.

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

## A   Extended History of Telecom Protocol Evolution

Protocols in wireless communication are the invisible contracts that let billions of radios share spectrum, identify one another, and negotiate Quality of Service (QoS). They differ from isolated hardware breakthroughs in that, once a rule set is adopted, every additional user or device deepens the network's value, creating a positive-feedback loop of interoperability, scale, and economic impact. From spark-gap transmitters to the emerging International Mobile Telecommunications-2030 (IMT-2030) vision, each protocol generation has arrived precisely when the previous rules could no longer unlock the next order of societal benefit.

From Marconi's spark-gap experiments in 1895 to the proliferation of Amplitude-Modulation and Frequency-Modulation (AM/FM) broadcasting in the 1930s, wireless links functioned as largely self-contained point-to-point systems. Message formats were improvised, spectrum was treated as private acreage, and interference was mitigated chiefly through geographic separation. A first, tentative move toward codified coordination came with the 1906 International Radiotelegraph Convention, which standardised station identifiers and adopted a universal distress call but offered little additional guidance; nonetheless, it proved that trans-border governance was feasible. For the next half-century, the ether remained virtually protocol-empty. A genuine system perspective emerged only after large-scale propagation campaigns—most notably those by Okumura—were distilled into closed-form path-loss expressions, and the subsequent 1980 formalisation of these data as the Hata model Hata [2013] provided engineers with a quantitative slide rule for planning frequency-reusable small cells. This analytical lens opened the way for dedicated cellular control channels and, ultimately, for the layered protocol stacks that characterise modern mobile networks.

**1G—Channelised Voice.**   Quantitative propagation models such as the Hata formulation provided the first system-level insight that radio spectrum could be reused aggressively Bernhardt [1987], and this analytic foundation precipitated the first generation (1G) of analogue cellular networks. Early systems—most notably the Advanced Mobile Phone System (AMPS) and the Nordic Mobile Telephone (NMT) network defined in Telecommunications Industry Association Standard (TIA) 553 MacDonald [1979] —shared a common control channel implemented as a narrow-band frequency-shift keying (FSK) stream carrying three message types: ORIGINATION, PAGE RESPONSE, and HANDOFF. This concise signalling grammar satisfied the era's central requirement of seamless city-wide mobility within tight frequency-reuse clusters and demonstrated that automated hand-off could accommodate a rapidly growing subscriber base Lee [1989].

The analogue architecture, however, exposed serious weaknesses. Conversations were transmitted as unencrypted FM audio, roaming identity was absent, and capacity was constrained by fixed guard bands that grew ever more expensive as handset density increased; eavesdropping required little more than a consumer scanner. These limitations drove the transition to digital second-generation (2G) protocols.

**2G—Digital Identity and Security.**

Digital second-generation (2G) systems were designed explicitly to eliminate the three vulnerabilities of the analogue era—clear-text audio, weak identity management, and rigid spectrum utilisation—and to support the rising expectation that mobile phones should work across national borders. The Global System for Mobile Communications (GSM) replaced frequency-modulated speech with full-rate and half-rate vocoders surrounded by convolutional coding and cyclic redundancy checks, then applied stream ciphers derived from a per-session challenge–response procedure Gerstlauer et al. [2000]. Subscriber credentials moved from handset firmware to a removable Subscriber Identity Module (SIM), enabling both secure roaming and mass-market prepaid services. A Time-Division Multiple-Access (TDMA) frame with eight slots compressed eight encrypted calls into the bandwidth that one analogue conversation had occupied, increasing spectral efficiency by nearly an order of magnitude. Interim Standard 95 (IS-95) pursued the same goals with direct-sequence Code-Division Multiple Access (CDMA), spreading each user's signal across the full carrier and achieving soft capacity that grew with signal-to-interference ratio. Although conceived only as control-plane text, GSM's mobile-originated short-message procedure was quickly commercialised as the 160-character Short Message Service (SMS), illustrating how richer signalling grammars could spawn unanticipated revenue streams and setting a precedent for the data-centric evolutions that would define the third generation.

**3G—State Machines for Soft Handover.**

Although 2G systems such as GSM and IS-95 multiplied spectral efficiency and introduced ciphered speech, they remained voice-centric, circuit-switched, and locked to kilobit-per-second data rates. The rapid uptake of laptops and early smartphones exposed those limits and motivated a new air interface capable of packet-switched megabit throughput. This requirement defined the third generation (3G) under the International Mobile Telecommunications-2000 (IMT-2000) umbrella.

The Universal Mobile Telecommunications System (UMTS) adopted Wideband Code Division Multiple Access (WCDMA) on a 5 MHz carrier and inserted a Radio Network Controller (RNC) between the base station and the core. Variable-rate convolutional and turbo coding, together with 1,500-Hz-cycle power-control commands, maintained link quality in dense urban multipath, while soft handover allowed a handset to combine energy from multiple cells Holma and Toskala [2005]. Because these techniques could not be expressed with the GSM Layer-3 grammar, the Third Generation Partnership Project specified a four-state Radio Resource Control (RRC) machine in TS 25.331 and introduced new primitives such as `MEASUREMENT_REPORT`, `ACTIVE_SET_UPDATE`, and high-frequency scheduling commands. Handover logic migrated to the handset, enabling data rates above 1 Mb/s and permitting applications to adapt radio requirements in real time. These protocol advances completed the transition from voice-first mobility to data-driven connectivity and set the stage for the orthogonal-frequency-division-multiple-access Long-Term Evolution (LTE) architecture that would follow in the fourth generation (4G).

**4G—Bearer Abstraction and All-IP Core.**

Even with WCDMA's megabit-class links, third-generation networks remained spectrum-constrained because spreading codes were finite and the uplink and downlink shared a coupled bandwidth that limited scheduling agility. These constraints set the stage for the 4G, formalised as Long-Term Evolution (LTE).

LTE replaced code-division multiplexing with Orthogonal Frequency-Division Multiple Access (OFDMA) on the downlink Yin and Alamouti [2006] and Single-Carrier Frequency-Division Multiple Access (SC-FDMA) on the uplink, while embracing multi-stream Multiple-Input Multiple-Output (MIMO) antenna processing Paulraj et al. [2004]. Together, these techniques tripled spectral efficiency without requiring additional spectrum and lifted peak user rates far beyond the reach of 3G. The architectural breakthrough, however, lay higher in the stack: the Evolved Packet System (EPS) bearer specified in 3GPP TS 36.300. Because both the control plane (using Diameter for policy) and the user plane (using GPRS Tunnelling Protocol–User, GTP-U) carried pure IP, application developers could count on predictable latency and bandwidth. These features, documented succinctly in Raj Jain's "Introduction to LTE" notes, enabled voice, video, and data to converge on an all-IP core and catalysed the mobile-app economy years before massive-MIMO hardware became commonplace.

**5G—Service-Based Architecture.**

While 4G LTE unified voice, video, and data on an all-IP core, it struggled to meet the emerging demands of ultra-low latency, massive IoT, and immersive applications. To address these limitations, 3GPP introduced the fifth generation (5G) with New Radio (NR), formalised in TR 38.913. It defined a tri-polar service model—enhanced Mobile Broadband (eMBB), Ultra-Reliable Low-Latency Communications (URLLC), and massive Machine-Type Communications (mMTC)—alongside flexible subcarrier numerologies and beam-centric massive MIMO Larsson et al. [2014].

Beyond the air interface, 5G's core network (TS 23.501/502) adopted a service-based architecture, replacing rigid LTE signalling with microservices exposed via HTTP/2 and JSON. Sessions now carry explicit QoS profiles tied to eMBB, URLLC, or mMTC needs, enabling true network slicing. Recent releases extend this model to non-terrestrial networks, Reconfigurable Intelligent Surfaces (RIS), and native AI integration, positioning 5G not just as a faster pipe but as a programmable spatial platform Huang et al. [2019].

**6G—Protocols for Distributed Cognition.**

Building on the service-centric flexibility of 5G, the sixth generation (6G) envisions a leap from connectivity to distributed cognition. The ITU-R IMT-2030 framework outlines a new protocol vocabulary to support this shift: symbol-level beam tracking for sub-terahertz channels, joint communication–sensing pilots that unify radar and data functions, and federated learning primitives such as `MODEL_REGISTER` and `GRADIENT_PUSH` for on-device intelligence coordination. While hardware

prototypes already demonstrate 100 Gb/s links at 140 GHz, such capabilities remain confined to the lab without standardised signalling to govern when, why, and how each node should act.

As with every generation before, physical breakthroughs alone are not enough. True scale and societal impact emerge only when those breakthroughs are encoded into shared protocols. From analogue control tones to JSON APIs and now AI-native message exchanges, wireless systems evolve by teaching radios to speak a richer language of coordination.

Seen in sequence, each protocol generation has delivered a specific new capability—mobility, digital security, mobile internet, all-IP convergence, service slicing, intelligent surfaces—that unlocked a fresh wave of economic and social value while postponing spectrum exhaustion. The lesson is clear: continued protocol research is not a peripheral activity but the core enabler of every subsequent hardware advance. Without new rule sets to orchestrate spectrum, topology, and compute, future breakthroughs in terahertz silicon, satellite constellations, or AI accelerators will remain islands of potential rather than the next shared infrastructure Letaief et al. [2019], Anthropic [2024], Azari et al. [2022].

