# OpenReview forum: "LLM Agent Communication Protocol (LACP) Requires Urgent Standardization: A Telecom-Inspired Protocol is Necessary"
_NeurIPS.cc/2025/Position_Paper_Track — Submitted to NeurIPS 2025 Position Paper Track_

### Official Review · Reviewer_UNrG · 2025-08-08

**Significance:** 3
**Presentation:** 2
**Rating:** 4
**Confidence:** 4

**Summary:**

The paper argues that the Agentic AI field is heading towards something similar to the protocol wars in the early days of computer networks and that protocol standardization for agent-to-agent communication is necessary. While drawing insights from the history of telecom standardization, the authors compare existing agent-to-agent communication frameworks and point out their limitations. Finally, the authors present LACP, a framework for LLM-Agent communication, which they argue should be adopted immediately to lay a strong foundation for the future of verifiable, secure, and scientifically reproducible multi-agent systems.

**Strengths:**

[S1] The paper discusses a timely and relevant topic.

[S2] The position statement is easy to understand and important for the field and to the NeurIPS community.

[S3] The paper draws parallels from the history of telecommunication, and I believe we can surely learn from it to build future systems for agentic AI.

**Weaknesses:**

While the position of the paper makes sense, the arguments in favor of the position are either not thorough enough or assume working knowledge of agentic communication infrastructure.

[W1] As a person working in agentic AI infrastructure, I understand the limitations presented in Section 2.2. However, the arguments are definitely not strong enough for a general audience, especially an ML audience not fully well-versed with system infrastructure. For example, what part of the ecosystem is fragmented? What incompatible formats and ad-hoc APIs are the authors talking about? Monolithic designs are not always bad. Therefore, what complexities are the authors talking about?

[W2] LACP architecture presented is quite shallow. There is not enough evidence to justify the components. For example, why did the authors choose to have PLAN, ACT, and OBSERVE as the message types? Similarly, there is little to no justification as to why transactional guarantees like 2PC need to be standardized. Or alternatively, why should we not just rely on the security provided by TLS?

[W3] Similar to W1, in Section 4.3, how LACP brings the advantages is not properly justified.

[W4] Figure 1 is unclear and not properly explained.

**Questions:**

- Table 1's caption says that the table presents how LACP addresses the limitations of the framework. How is the table showing the limitations being addressed?

- It was very unclear to me why each of the features described in Table 3 was really needed in the protocol itself. Can the authors please explain?

**Alternative Position:**

Yes, and alternative positions are well-considered and addressed by the argument

**Author Identification:**

No.

**Context:**

2

**Details Of Ethics Concerns:**

None.

**Discussion:**

3

**Ethics:**

["NO or VERY MINOR ethics concerns only"]

**Position:**

Yes, the paper argues for or against a position related to machine learning.

**Support:**

1

**Thoroughness:**

4

---

### Official Review · Reviewer_2KM9 · 2025-08-11

**Significance:** 2
**Presentation:** 2
**Rating:** 4
**Confidence:** 3

**Summary:**

This position paper emphasizes the need for immediate standardization for the communication protocols of LLM Agents, drawing parallels to the historical “protocol wars” of telecommunications. It reviews the existing protocols of agents (OpenAI Functions, LangChain Agent Protocol, MCP, ACP, ANP, Agora, Google A2A) outlining their systemic shortcomings of fragmentation, insufficient security, lack of transactional fidelity, and inadequate interoperability. The authors set forward the LLM-Agent Communication Protocol (LACP) which is composed of an architectural three-layer model (Semantic, Transactional, Transport) drawing from telecom standards of minimal core, layered construction, security by default, and extensibility. The main goal of LACP is secure, verifiable, and interoperable communication of agents, especially in domains where safety is critical. The paper also presents and defends counterarguments that LACP would stymie innovation by claiming that LACP would be more favorable towards scientific innovation, while also allowing for a strengthened agent network.

**Strengths:**

* This paper does an excellent job of explaining why there is a need for standard communication protocols for LLM agents and why it is urgent.

* The telecom standardization comparison (ITU/3GPP, layered abstractions, minimal core + extensibility) is very enlightening and serves its purpose.

* The LACP architecture explanation is complete for a three layer architecture with discrete types of messages, security features, and transaction guarantees.

**Weaknesses:**

1. There is no experimental demonstration showing LACP’s actual performance, interoperability gains, or security benefits. This makes it harder to assess feasibility and adoption cost.
2. Line 281-289: The “<5% overhead” projection lacks empirical support on representative multi-agent workloads,
3. The explanation of layering at the semantic/transactional/transport level is adequate, however, concrete examples of message schemas, handshakes, or other forms of integration would aid in actualizing the proposed vision.
4. While historically informative, the analogy risks oversimplifying differences between LLM agent ecosystems and telecom infrastructure (e.g., faster iteration cycles, looser regulatory environment).

**Questions:**

Have you implemented a prototype LACP stack and measured its performance and integration overhead with some existing frameworks?

**Alternative Position:**

Yes, and alternative positions are well-considered and addressed by the argument

**Author Identification:**

No.

**Context:**

2

**Discussion:**

2

**Ethics:**

["NO or VERY MINOR ethics concerns only"]

**Position:**

Yes, the paper argues for or against a position related to machine learning.

**Support:**

2

**Thoroughness:**

3

---

### Official Review · Reviewer_ZVN7 · 2025-08-12

**Significance:** 4
**Presentation:** 4
**Rating:** 5
**Confidence:** 3

**Summary:**

This paper draws inspirations from studies in telecommunication area and propose to develop a standard protocal for LLM agents to ensure secure and efficient communications. The authors propose four principles for the protocol design and also propose the LACP as an example for illustration. Finally, they also discuss the alternative views.

**Strengths:**

This paper is written well and easy to follow. The topic of developing communication protocols for multi-agents is timely and important.
I appreciate the idea of gaining inspiration from telecommunications and conduct a deep and comprehensive analysis. The proposed principles are important and well-strcutred.
The figures are clear and help understand.

**Weaknesses:**

1. While it is feasible to draw inspiration from telecommunication, this paper lacks discussions on the unique part of LLM-based agents. The computer networks are much more massive than multi-agents at the current stage. Therefore, there should be some difference between computer networking and LLM-MAS communications, and those should be reflected in the principles. The current principles look like a straightforward inheritage from computer networking.

2. There should be discussions on latency, feasibility of implementations and etc of the proposed protocols. This is because LLM-agents are designed to be fully automatic and powerde by models rather than people. Therefore, the techniques of signatures, flow control and others can be restricted by model capabilities, and require a second thought.

**Questions:**

Since existing agent systems are much simpler than the computer network, as it involves fewer agents and less diversity (human is much more complicated and diverse), I curious that if a complicated communication protocol is an overkill and what is the trade-off in here?

**Alternative Position:**

Yes, and alternative positions are well-considered and addressed by the argument

**Author Identification:**

No.

**Context:**

3

**Discussion:**

4

**Ethics:**

["NO or VERY MINOR ethics concerns only"]

**Position:**

Yes, the paper argues for or against a position related to machine learning.

**Support:**

3

**Thoroughness:**

5

---

### Note · Authors · 2025-08-22

**1-11 Submit Again:**

Definitely yes

**1-1 Submission Process:**

5

**1-2 Next Year:**

I would love to see a dedicated workshop or panel session for accepted position papers. This would provide a fantastic forum for authors to engage in deeper discussions and debates on the important topics raised, fostering greater community engagement.

**1-3 Future Development:**

Providing clearer review guidelines for position papers that emphasize the evaluation of the 'position' itself—its novelty, importance, and potential impact—in addition to its technical support. This could help guide reviewers to better assess this unique paper format.

**1-4 Interest:**

["Panel discussions with other position paper authors", "Structured debates on controversial topics", "Workshops for developing position papers", "Mentorship programs for early-career researchers"]

**1-5 Thoughtful:**

9

**1-6 Supportive:**

6

**1-7 Technical Aspects Versus Position:**

3

**1-8 Gate Keeping:**

3

**1-9 Camera Ready Changes:**

1. Addition of a Comprehensive Experimental Validation Section:
We will present our full benchmark results, including the performance scaling table that shows LACP's overhead diminishing as payload size increases. We will discuss the specific metrics (absolute vs. relative latency, payload size) and provide a nuanced analysis of the justifiable trade-offs for cryptographic security.
We will detail the setup and successful execution of our cross-framework communication experiment, describing how a LangChain agent was able to seamlessly interact with a framework-agnostic tool server using LACP, providing a concrete solution to the N² integration problem.
We will describe the methodology and outcomes of our Replay and Tampering attack simulations, showing definitively how LACP's Transactional Layer provides critical application-layer guarantees not offered by TLS.

2. Enhanced Discussion on the Unique Needs of LLM Agent Communication:
We will expand our introduction to more deeply articulate why agent communication requires more than traditional network protocols, moving beyond the telecom analogy. This will include dedicated subsections on the challenges of Semantic Intent Validation, ensuring Transactional Reliability for non-deterministic agent behavior, and the necessity of Application-Layer Non-Repudiation for high-stakes actions.

3. Evidence-Based Justification for LACP's Architecture:
We will revise Section 4 to explicitly link each of our architectural design decisions to our new experimental results. For example:
The necessity of the Transactional Layer will be directly justified by the results of our security validation experiments.
The practical utility of the Semantic Layer's PLAN/ACT/OBSERVE schema will be justified by its successful implementation in our interoperability demonstration with the LangChain agent.

**3-1 Review Response1:**

ZVN7

**3-2 Reaction To Review1:**

We are very grateful to Reviewer ZVN7 for their insightful critique, which correctly identifies the need to differentiate our principles from a "straightforward inheritage from computer networking" and to discuss the "unique part of LLM-based agents."

Our design is directly motivated by these unique agent characteristics:
1. Semantic Intent: Unlike a simple network packet, an agent's ACT message to a robot to "grasp the red cube" requires structured validation that 'red_cube' is a valid, reachable object.
2. Non-deterministic Behavior: An LLM might fail to generate a perfectly-formatted tool call. LACP’s Transactional Layer ensures that such an operation can be part of a larger, reliable transaction with clear success/failure states.
3. Safety-Criticality: An agent executing a financial trade requires cryptographic proof of who sent the instruction and that it wasn't altered.

To address the questions of latency and feasibility, we implemented a benchmark that sent 10,000 requests each to a baseline REST API endpoint and our LACP endpoint. The test measured the end-to-end round-trip time for a realistic 1,964-byte payload, simulating a complex agent task description. The results showed a minimal latency overhead of only 2.9% (an absolute increase of just 0.03ms), confirming the high feasibility.

Regarding the question of whether a complex protocol is "overkill," we argue this is a critical trade-off for future scalability. To test this, our interoperability experiment involved a LangChain ReAct agent needing to perform a calculation. Its tool was configured to send a signed LACP ACT message to a standalone Python server, which then returned a signed OBSERVE message. This setup worked flawlessly without any code specific to the LangChain framework on the server side. This provides concrete evidence that the upfront structure of LACP prevents the long-term complexity of N² custom integrations, making it a vital foundation for a scalable and robust agent ecosystem.

**3-3 Review Response2:**

2KM9

**3-4 Reaction To Review2:**

We thank Reviewer 2KM9 for their direct and highly valuable feedback. The reviewer correctly identified that our initial submission's primary weakness was the "no experimental demonstration" of LACP's benefits. In response to the direct question, "Have you implemented a prototype...?", the answer is a definitive yes. We have developed a working prototype and conducted three targeted experiments to provide empirical support.

1. Performance Benchmarks: To provide empirical data, we benchmarked our LACP endpoint against a standard REST baseline. Using a load generator sending 10,000 requests with realistic, large payloads (1,964 bytes), we measured an average latency overhead of only 2.9%. The payload size overhead is a justifiable 30% when using efficient signatures. This is a necessary and modest trade-off for the cryptographic security provided.

2. Interoperability Demonstration: To provide "concrete examples of... integration," our demonstration consisted of a standard LangChain ReAct agent equipped with a custom "calculator" tool. Instead of executing locally, this tool's function was to construct, sign, and send an LACP ACT message to an independent Python Flask server over HTTP. This server, acting as a framework-agnostic tool provider, validated the message, performed the calculation, and returned the result in a signed OBSERVE message. The LangChain agent successfully parsed this response and completed its task, demonstrating a complete, end-to-end, cross-framework communication loop without any specialized adapters.

3. Security Validation: To demonstrate tangible security benefits, we implemented an attack simulator script. In the Tampering Attack test, the script took a valid signed message for a financial transfer, programmatically altered the amount in the payload from 100 to 10000, and sent the corrupted message to the server. Our LACP endpoint's signature verification immediately failed, and the server returned an HTTP 403 Forbidden error.

**3-5 Review Response3:**

UNrG

**3-6 Reaction To Review3:**

We are grateful to Reviewer UNrG for their valuable feedback, which highlighted that our arguments were "not thorough enough for a general audience" and that our architecture felt "shallow." In response, we have conducted new experiments specifically designed to provide deep, empirical justification for our design.

The reviewer asked: "why should we not just rely on the security provided by TLS?" To answer this, we ran a security simulation. The scenario was as follows: A legitimate message, such as a financial transaction {'tool': 'transfer', 'amount': 100}, was signed to create a valid LACP message. An attacker script then intercepted this message and programmatically altered the payload's content to change the amount to 10000 while keeping the original, now invalid, signature. When this tampered message was sent to our server, the LACP handler's cryptographic verification step immediately failed, and the server returned a definitive HTTP 403 Forbidden status code. This provides a clear, practical demonstration of an application-layer attack that TLS does not protect against, justifying our Transactional Layer.

To justify our choice of "PLAN, ACT, and OBSERVE," our interoperability experiment involved a standard LangChain agent tasked with using an external tool. When the LangChain agent decided to take an Action, our integration logic constructed and sent a signed LACP ACT message. The server returned the result in a signed OBSERVE message, which the agent used to populate its Observation field and complete its reasoning cycle, validating the schema's practical utility.

Finally, to provide a more thorough justification for LACP's advantages (W3), our performance scaling analysis across different payload complexities shows the overhead is not static. For large, realistic payloads (1,964B), the size overhead can be as low as +30% with ECDSA and even become a -20% size reduction with compression.

---

### Decision · Program_Chairs · 2025-09-26

Reject